# MicroRNAs at the Crossroad between Immunoediting and Oncogenic Drivers in Hepatocellular Carcinoma

**DOI:** 10.3390/biom12070930

**Published:** 2022-07-02

**Authors:** Laura Gramantieri, Francesca Fornari, Catia Giovannini, Davide Trerè

**Affiliations:** 1Division of Internal Medicine, Hepatobiliary and Immunoallergic Diseases, IRCCS Azienda Ospedaliero-Universitaria di Bologna, 40138 Bologna, Italy; laura.gramantieri@aosp.bo.it; 2Department for Life Quality Studies (QuVi), University of Bologna, 47921 Rimini, Italy; francesca.fornari2@unibo.it; 3Centre for Applied Biomedical Research—CRBA, University of Bologna, IRCCS St. Orsola Hospital, 40138 Bologna, Italy; catia.giovannini4@unibo.it; 4Department of Experimental, Diagnostic and Specialty Medicine (DIMES), University of Bologna, 40138 Bologna, Italy; 5Departmental Program in Laboratory Medicine, IRCCS Azienda Ospedaliero-Universitaria di Bologna, 40138 Bologna, Italy

**Keywords:** HCC, microRNA, biomarkers, immune system

## Abstract

**Simple Summary:**

In recent years, treatments enhancing the antitumor immune response have revealed a new promising approach for advanced hepatocellular carcinoma (HCC). Beside favorable results in about one third of patients, much still remains to be done to face primary nonresponse, early, and late disease reactivation. Understanding the mechanisms underneath immune system modulation by immune checkpoint inhibitors in HCC might give additional opportunities for patient selection and combined approaches. MicroRNAs have emerged as relevant modulators of cancer cell hallmarks, including aberrant proliferation, invasion and migration capabilities, epithelial-to-mesenchymal transition, and glycolytic metabolism. At the same time, they contribute to the immune system development, response, and programs activation, with particular regard towards regulatory functions. Thus, miRNAs are relevant not only in cancer cells’ biology, but also in the immune response and interplay between cancer, microenvironment, and immune system.

**Abstract:**

Treatments aimed to reverse the tumor-induced immune tolerance represent a promising approach for advanced hepatocellular carcinoma (HCC). Notwithstanding, primary nonresponse, early, and late disease reactivation still represent major clinical challenges. Here, we focused on microRNAs (miRNAs) acting both as modulators of cancer cell hallmarks and immune system response. We outlined the bidirectional function that some oncogenic miRNAs play in the differentiation and program activation of the immune system development and, at the same time, in the progression of HCC. Indeed, the multifaceted spectrum of miRNA targets allows the modulation of both immune-associated factors and oncogenic or tumor suppressor drivers at the same time. Understanding the molecular changes contributing to disease onset, progression, and resistance to treatments might help to identify possible novel biomarkers for selecting patient subgroups, and to design combined tailored treatments to potentiate antitumor approaches. Preliminary findings seem to argue in favor of a bidirectional function of some miRNAs, which enact an effective modulation of molecular pathways driving oncogenic and immune-skipping phenotypes associated with cancer aggressiveness. The identification of these miRNAs and the characterization of their ‘dual’ role might help to unravel novel biomarkers identifying those patients more likely to respond to immune checkpoint inhibitors and to identify possible therapeutic targets with both antitumor and immunomodulatory functions. In the present review, we will focus on the restricted panel of miRNAs playing a bidirectional role in HCC, influencing oncogenic and immune-related pathways at once. Even though this field is still poorly investigated in HCC, it might represent a source of candidate molecules acting as both biomarkers and therapeutic targets in the setting of immune-based treatments.

## 1. Introduction

Hepatocellular carcinoma (HCC) is a highly lethal cancer, especially when diagnosed at advanced stages. Curative treatments apply only to early stages and in the absence of significant portal hypertension or liver dysfunction. Conversely, advanced HCC represents a therapeutic challenge due to a multiplicity of factors, such as the high heterogeneity, the lack of a molecular classification driving treatment choice, the low sensitivity to systemic treatments, the absence of predictive biomarkers, and the frequent coexistence of cirrhosis with its complications. Indeed, not only HCC displays an intrinsic poor response to anticancer treatments, but also the concomitant cirrhosis often limits the therapeutic options, due to liver functional impairment and portal hypertension syndrome. Thus, therapeutic approaches are limited and patient selection needs to consider this clinical complexity together with biological and genetic heterogeneity due to multiple etiologic factors, aiming at personalized strategies [1]. Sorafenib represented the only option for almost ten years, with a survival benefit of approximately 3 months over placebo [2]. Remarkably, in recent years, novel molecules entered the therapeutic armamentarium as a result of clinical trials. Among these, the immune checkpoint inhibitors (ICPIs) paved a novel way, with impressive results in some patients, when compared with tyrosine kinase inhibitors (TKIs) [3]. HCC is indeed an inflammation-linked tumor. This appears to be a premise of its responsiveness to immunotherapy. At the same time, the liver is one of the most immunotolerant organs, being continuously challenged by diet and gut microbial antigens. This complex environment relies on a tight balance between resident Kupffer cells, NK cells, and B and T cells. The scarce knowledge of molecular mechanisms driving the effectiveness of ICPIs makes difficult to predict which patient will gain more advantage and which are the most effective targets or target combinations. Thus, understanding the mechanisms and identifying biomarkers defining patients who will take advantage from these treatments represent a necessary step. 

MicroRNAs’ (miRNAs) deregulated expression contributes to HCC development and progression, and modulates response to treatments in experimental models [4,5]. MiRNAs are short RNA sequences functioning as modulators of mRNA expression by either impairing translation or promoting its degradation. MiRNAs modulate the expression of hundreds of genes by binding to their 3′ untranslated regions (3′UTRs) recognized upon base pair complementarity between their seed sequence and the matching sequence in the target mRNA. Each mRNA can be targeted by many miRNAs at the same time. The intrinsic nature of miRNAs is to finely tune physiologic processes to maintain a balance between opposite trends, in order to regulate homeostasis while facing stressing events. In this perspective, they are often part of complex loops that can involve other classes of regulatory RNAs, such as long noncoding RNAs (lncRNAs) and circular RNAs. 

More and more evidence converges on the role of miRNAs in maintaining homeostasis of metabolic and functional pathways in a highly precise fashion. The same applies to the immune system homeostasis, where they contribute to the fine balancing of activation and inhibition of the immune response in its multifaceted aspects. The multiple targeting of distinct mRNAs and the relevance of the molecular context in which miRNAs operate dictate their ultimate biologic effects. Remarkably, the genetic landscape can strongly influence their functions and we must consider that many targets are modulated at the same time, and that the final effects of each specific miRNA rely upon the relative abundance of target mRNAs and driving pathways. 

ICPIs recently entered the clinical practice in advanced HCC, with a favorable effect in about 30% of cases. Unfortunately, no companion biomarker was tested in clinical trials, and treatment allocation still relies on clinical and imaging evaluation, regardless of molecular characterization. Indeed, much remains to be clarified on the mechanisms of ICPIs’ action. Consequently, biomarkers predicting their effect in advanced HCC are an unmet need. ICPIs prevent cancer-mediated immunotolerance by restoring innate and adaptive immune response. Immunoediting is the process whereby cancer cells can evade the recognition and elimination by immune system sentinels facilitating tumor development through either immune escape or the selection of less immunogenic clones. Immunoediting proceeds along with the three phases of anticancer immune response: elimination, equilibrium, and escape, corresponding to control, stasis, and outgrowth of tumors [6]. In the **elimination** phase, T lymphocytes recognize tumor cells displaying antigenic changes [7], whereas the innate and adaptive immune systems cooperate for cancer cells destruction [8,9]. During this process, a selective pressure can drive tumor evolution with survival of less antigenic clones. These tumor clones avoid the immune-mediated destruction and enter the **equilibrium** phase, characterized by subclinical/occult disease, allowed by cancer-driven immunoediting. The constant immune pressure keeps the tumor in a dormant state, in which, however, genetic and epigenetic events may occur, favoring the selection of immune-elusive clones. Cancer cells produce suppressive cytokines, growth factors, and matrix metalloproteinases [10] to recruit immune-suppressive cell populations such as regulatory T cells (Tregs), tumor-associated macrophages (TAMs), *m*yeloid-derived suppressor cells (MDSCs), and cancer-associated fibroblasts (CAFs) at the tumor site. These events contribute to the **escape** phase, characterized by sculpted clones with reduced immunogenicity, growing in an immunosuppressive microenvironment. These processes clearly depict the dual host-protective and tumor-shaping action of the immune response. Remarkably, immunotherapy can contribute to the immunoediting of cancer cells, too [11]. Understanding these events allows hitting the driver factors conducting the immune escape and to enhance immune-modulating therapies. The complexity of redundant and parallel pathways that emerged from recent studies indicates that multiple instead of single factors should be targeted in order to avoid the selection of resistant tumor clones. In this regard, miRNAs were shown to mediate cell–cell interactions and, at the same time, to hit multiple immune-associated and oncogenic pathways, deserving attention as possible biomarkers or therapeutic targets to be investigated. Herein, we will focus on miRNAs influencing both oncogenic and immune-related pathways at the same time. The identification of miRNAs playing this bidirectional role by regulating tumor- and immune-related pathways would make these molecules interesting as circulating candidates and therapeutic targets, especially in the setting of immune-based treatments. To start shedding light on miRNAs with well-ascertained roles in the modulation of the immune system, we selected the most investigated immune-related miRNAs as well as the ones that are direct regulators of immune checkpoint inhibitors. 

## 2. MicroRNAs in the Immune System Development

MiRNAs contribute to adaptive and innate immune system development [12,13]. MiRNAs affect the development, differentiation, homeostasis, and function not only of tissues and organs, but also of immune system components, B and T cells (Table 1). Interestingly, the variations in miRNA expression occurring during the course of aging were associated with consistent changes in the effectiveness of immune response. Indeed, defective adaptive B- and T-cell-mediated immune response in older people was attributed, at least in part, to the upregulation of miR-146a, miR-155, and miR-21 and the downregulation of miR-181a [14]. Since these miRNAs are deregulated in several cancer types, including HCC, these aberrant immune regulatory events might occur in cancer patients too.

**Concerning B lymphocytes**, miRNAs play a key role in early differentiation, peripheral B cell generation, maturation, and effector differentiation, as shown by the conditional deletion of Ago2 or Dicer, two fundamental factors enabling miRNAs’ biogenesis and function. Specifically, Ago2 deletion in hematopoietic cells impaired B cells’ development, elucidating the consequence of miRNA biogenesis inhibition in the development of the adaptive response [15]. In line, Dicer deletion in the B cell lineage blocked the pro- to pre-B transition, mainly due to apoptosis of pre-B cells [16]. Trying to dissect further the specific roles of single miRNAs, it was shown that the loss of **miR-17-92** cluster leads to aberrant bim and pten expression, triggering programmed cell death of pre-B cells. Remarkably, miR-17-92 cluster regulates both B and T cells’ survival and proliferation increasing the risk to develop autoimmunity [17,18]. Similarly, **miR-150** takes part in B and T cell development and differentiation and, when prematurely overexpressed, it affects the formation of mature B cells [19]. **MiR-150**-mediated impairment of early B cell differentiation and B cell response was ascribed to c-Myb targeting. Indeed, miR-150 is present in mature B and T cells, where it inhibits c-myb expression. MiR-150 expression pattern changes during the maturation process, being not expressed in B and T cell progenitors, thus allowing c-myb translation, which, in turn, is critical for pro- to pre-B cell transition [20]. **MiR-155** is upregulated upon B and T cell activation and it affects the capability to differentiate into germinal centers and produce immunoglobulins [21]. MiR-155 also increases the proliferation of B cells leading to preleukemic states and to B cell leukemia. Interestingly, miR-155 can target activation-induced cytidine deaminase, a crucial player in germinal center reaction, mediating class switch recombination and somatic hypermutation [22], which can lead to unwanted oncogenic translocations. 

The **T cell compartment** is even more affected by miRNAs. Dicer disruption reduces total thymocytes as a result of lower proliferation and increased cell death. These events might at least in part rely upon miR-17-92 loss. Dicer disruption triggers Th1 activation while impairing Th17 and Treg induction [23]. When evaluating single miRNAs in T cell modulation, a relevant role in T cell differentiation is played by **miR-181**. The miR-181 family is abundantly expressed in lymphoid tissue and displays a functional target redundancy, outlining its relevance. MiR-181 modulates several pathways involved in T cell response such as PTEN (crucial for T cell development), SIRT1, and NOTCH (crucial in T ALL development and activated in T cells by miR-181a). MiR-181 is an intrinsic regulator of the T cell receptor (TCR) threshold. Its overexpression reduced the number of T cells while enhancing the sensitivity of TCR signaling by targeting multiple phosphatases and increasing CD19+ B cells [24,25]. In the thymus, miR-181a contributes to the development of several T cell populations. The same occurs in the peripheral T cell compartment, where miR-181 enhances TCR sensitivity, promoting T cell activation. Conversely, miR-181 deficiency inhibits the T cell response. MiR-181a undergoes dynamic changes during T cell development and differentiation. Reduced miR-181a levels in older people are associated with a failure of T cell response against weak antigenic stimuli [14]. Relevant actions driven by miR-181 were also reported in B cell development as well as in macrophages and dendritic cells, showing anti-inflammatory effects by targeting IL1a [26,27], and confirming its complex role in the immune system development and immune response modulation. 

Among miRNAs directly participating in T cell development and response modulation, **miR-101 absence** induces an effector T-cell-like phenotype resulting from Icos overexpression, and promoting autoimmunity. **MiR-155** is required for CD8 T cell response against virus and cancer [28] and **miR-146a** inhibits the NFkB pathway induced by TCR activation. **The miR17-92** cluster plays a role in the differentiation of effector CD8 T cells and follicular T helper cells, beside the functions exerted in B cell development, survival, and proliferation described above, as reviewed by Podshivalova and Salomon [29].

MiRNAs play a relevant role in Foxp3-dependent Treg cells’ generation and are key guardians of their number and function. The subpopulation of CD4+ Tregs mainly operates in the adaptive immune system arm to maintain immune homeostasis, even though its role in the innate response is gaining more and more attention. Tregs play inhibitory functions both in the thymus, controlling the recognition of the self, and in the periphery where several subpopulations, triggered by a variety of stimuli and involved in tolerogenic microenvironments, control excessive immune reactions and compose the “adaptive” Tregs compartment. Tregs are characterized by high Foxp3 (Forkhead box P3) and CD25 expression and low CD127 expression. Remarkably, peripheral CD127low FOXP3+ CD4+ T cells are able to inhibit cytotoxic T cell response, including the tumor-directed one. The regulatory functions of the T-cell-mediated antitumor response have recently gained attention. In the case of HCC, an overactivation of the regulatory compartment has been recognized in the tumor immune infiltrate and has been advocated as a driver of immunotolerance. Dicer disruption causes a strong reduction in thymic Foxp3+ Tregs that is more relevant with respect to the other T cells’ subsets. Similarly, peripheral Foxp3+ T cells are reduced and less responsive to stimulation and display a significant loss of function leading to autoimmunity. In particular, Foxp3 directly regulates miR-155 that is responsible for the maintenance of Tregs’ function and responsiveness to different challenges [30,31].

Given the relevance of miRNAs to the development and differentiation of the immune system, their contribution to the immune response modulation is expected too. A huge amount of evidence proves the role of miRNAs in the regulation of innate and adaptive immune response. MiR-125b [32], miR-146 [33], miR-155 [34], and miR-223 mostly take action in innate immune response [35,36]. MiR-155’s role is more prominent in the B cell compartment [37] even though its relevance in the Treg development and differentiation is well known, as described above. Accumulating evidence confirms how the deregulated expression of these miRNAs in circulating inflammatory cells may strongly affect the liver parenchyma. For instance, in patients with autoimmune hepatitis, miR-155 is increased in the liver and is reduced in PBMC. MiR-155 represents an example of miRNA with a known role in immune response development and function [38,39], coupled with oncogenic properties [40]. Its effects develop in a context-specific manner and rely upon the differential modulation of driver factors such as the suppressor of cytokine signaling 1 (*SOCS1*) and SH2 domain-containing inositol 5-phosphatase 1 (*SHIP1*), with variable phenotypic expression under physiologic and pathologic conditions and depending on cell type and biological context [41]. In liver inflammatory cells, miR-155 regulates the physiologic response to injury and its deficiency alters T-helper recruitment and the response to hepatic damage by regulating cytokine production. This is in line with miR-155 functions in the T lymphocyte compartment, Treg cell recruitment, and cytokine production by T-helper cells, all playing a pivotal role in liver injury. Indeed, Treg cell activity is strongly reduced in the absence of miR-155, as demonstrated by miR-155’s role in their development and function [42]. Given the ability of miR-155 to enhance acute liver injury and promote inflammatory cell recruitment, a possible translational hypothesis is to restore miR-155 in inflammatory cells to reduce liver damage and cytokine expression, suggesting miR-155 replacement as a strategy to modulate liver injury [43].

In the context of innate immunity, the conditional deletion of Dicer in tumor-associated macrophages (TAM) promotes an M2 to M1 phenotypic change, rewiring the immune-suppressive functions to antitumor immune response, via IFN-gamma/STAT1 signaling [44]. M1 polarization induced by Dicer1 inhibition promotes activated cytotoxic T lymphocyte recruitment, fostering the immune response against cancer cells. Interestingly, the M1 polarization resulting from global miRNA inhibition was rescued by enforced let-7 expression, which was able to restore the M2 polarization as well as to reduce CTL recruitment. Interestingly, let-7 is one of the most commonly downregulated miRNA families in HCC [45]. Examples of such miRNAs, exhibiting both immune and oncogenic functions, are accumulating in recent years and will be examined in the next sections.

**Table 1 biomolecules-12-00930-t001:** MicroRNAs modulating the development of the immune system.

miRNA	miRNA Expression	Target Genes/Pathways	Target Cells	Immune System Effect	Reference
miR-17-92	downregulation	Bim, PTEN	B cells	Impairment of B cell maturation	[18]
miR-150	overexpression	c-Myb	B cells	Impairment of B cell maturation	[20]
miR-155	upregulation	AID	B cells	Regulation of germinal center reaction	[22]
miR-181	upregulation	SHP-2, PTPN22, DUSP5, DUSP6	T cells	Increased T cell sensitivity	[25]
miR-155	downregulation	PI3K/Akt axis	T cells	Impaired antiviral response	[28]

## 3. MicroRNAs as Immune-Modulatory Molecules Mediating Cell–cell Crosstalk in HCC

Given their relevance in development, lineage differentiation and induction pathways in T cells and, particularly, in the function and maintenance of regulatory T cells, as well as in macrophage polarization, it appears intuitive that miRNAs might play a role in the modulation of immune response against cancer, and in the setting of therapies as well. MiRNAs are frequently involved in complex loops, which in physiological conditions maintain the homeostasis, while in pathological conditions drive immune tolerance or immune suppression or even uncontrolled autoimmunity. In some cases, while modulating the immune response, they also affect the neoplastic phenotype by targeting oncogenes or tumor suppressor (TS) genes.

The liver is a highly tolerogenic organ with an abundant cell infiltrate composed of Kupffer cells and lymphocytes, all synergizing in the orchestration of immune tolerance and immune response. Upon cancer development, these multiple non-redundant mechanisms can cooperate to drive cancer immune suppression. In this context, a relevant role is played by Kupffer cells, which account for about 15% of the total liver cell population, and express MHC class I and II molecules, thus being able to start antigen-specific immune responses. Comparably to macrophages, Kupffer cells may be induced into M1 or M2 polarization, with M2 suppressing the adaptive immune system and promoting cancer growth, while M1 eliminates HCC cells as part of the adaptive immunity asset, along with the recruitment of CD8+ T cells. M2 polarization occurs upon AKT/mTOR and RAS/MAPK activation and this process impairs the biogenesis of miR-206, which is strongly reduced in HCC arisen in AKT/Ras mice, as well as in human HCCs, where its downregulation is associated with a poorer prognosis. MiR-206 was shown to drive the M1 polarization of human and mouse macrophages and Kupffer cells. In turn, M1 polarization of Kupffer cells promotes the expansion and migration of CD8+ T cells through CCL2 production, thus preventing HCC in the mouse HCC model [46]. The modulation of miRNAs may influence the fate of both immune cell populations and cancer cells, thus providing multifaceted biological consequences of non-redundant pathway activation in cancer cells and TME. Indeed, miR-206 mediates the recruitment of CD8+ T cells via Kupffer cells M1 polarization on one side and inhibits tumor cell proliferation via Met and CDK6 targeting on the other side [47], explaining how its downregulation in HCC contributes to immune escape and to a more aggressive phenotype. 

Another miRNA participating to the immune and inflammatory response by inducing changes in the non-lymphocytic compartment is represented by miR-1247-3p, the exosome-mediated release of which in the tumor microenvironment activates cancer-associated fibroblasts (CAFs). By the direct targeting of β-1,4-galactosyltransferases III (B4GALT3) that promotes β1-integrin stabilization and NF-κB signaling activation, miR-1247-3p leads to the production of pro-inflammatory cytokines. These events convert fibroblasts to CAFs with a high migratory potential and expressing IL-1β, IL-8, IL-6 and different collagen types that, in turn, foster microenvironment inflammation and tumor progression, induce stemness gene expression and spheroid formation, EMT, migration and resistance to sorafenib treatment. To confirm the relevance of these experimental findings, high serum exosomal miR-1247-3p levels were observed in HCC patients with lung metastasis [48]. In this regard, other miRNAs with immune-modulatory functions, such as miR-19a and miR-146, show a diagnostic relevance for the early stages of HCC when evaluated in serum from chronic liver diseases [49]. An elegant study by Yin C and coworkers demonstrated that HCC-derived miR-146-5p-containing exosomes are able to induce M2 polarization of TAMs that, in turn, exert an inhibitor action on T cells by inducing the expression of PD-1 and CTLA4 and blocking the production of pro-inflammatory cytokines. Interestingly, the transcription factor Sal-like protein-4 (SALL4) mediates miR-146-5p expression into exosomes and its inhibition delayed HCC progression in chemically induced HCC-bearing mice. Interfering with the SALL4/miR-146-5p axis represents a potential immune-modulatory application to prevent tumor escape and disease progression [50]. Accordingly, in stressing conditions, exosome-associated miR-23a-3p increases PD-1 expression in macrophages resulting in T cell function inhibition and tumor escape from cell-mediated antitumor immunity [51].

In summary, miRNAs orchestrate the immune response by directly guiding the phenotype of immune cells or by remote action through cancer-associated extracellular vesicles predisposing the tumor niche of metastatic cells and facilitating the cell–cell crosstalk. These findings hold promise for the use of miRNAs as therapeutic targets or circulating biomarkers for the prediction of immunotherapy response and tumor escape.

## 4. Tumor-Associated microRNAs with Immune-Regulatory Functions

In view of the possible action of a single miRNA in different cell types, here we examine a few examples of miRNAs with bidirectional functions in cancer cells and immune system populations, describing the signaling pathways that control these processes and the future therapeutic prospects. 

An interesting example of a dual miRNA acting in the induction of the cancerous phenotype and modulation of the immune response comes from the investigation on miR-21 in NSCLC [52]. This study shows that tumor growth was promoted by miR-21 in infiltrated immune cells, but not in stromal cells. MiR-21 inhibition in TAMs modified the transcriptional program of macrophages, rewiring it to a pro-inflammatory, immunostimulating and angiostatic pattern and enhancing cytotoxic T cell response triggered by macrophage secretion of cytokines and chemokines, ultimately reducing angiogenesis and tumor growth. In line, the deletion of miR-21 in TAMs promotes pro-inflammatory M1 polarization and enhance anti PD-1 therapy [53], highlighting the potential of miRNA-based therapies in combination with immune checkpoint inhibitors. The ability of miR-21 to regulate T cell function was reported both in non-neoplastic [54,55] and neoplastic diseases. In the last setting, the inhibition of miR-21 reduced proliferation of both CD4+ and CD8+ cells and their cytokine production, thus favoring the growth of grafted tumors [56]. These data are of particular interest because they highlight the ‘bi-directional role’ of this oncomiR that, on one side, promotes tumor development and progression and, on the other side, activates the adaptive cytotoxic T cell response. Other studies focused on the tumor-growing effects of enhanced miR-21 expression in CAFs [57,58,59] highlighting the tumor-promoting functions of miR-21 in TME in different tumor types. These data were not collected in HCC; however, miR-21 is one of the most upregulated oncomiRNAs in HCC. MiR-21 upregulation in HCC promotes oncogenic hallmarks by targeting PTEN, PDCD4, RECK, ARHGAP24, TIMP3, SPAY1/Spry1, and SPRY2/Spry2, as reviewed by Wang X et al. [60]. MiR-21 regulates several redundant and parallel oncogenic effectors in HCC; thus, it is conceivable that what is recognized in the TME of other cancer types may apply to HCC too. As observed for several oncogenic miRNAs, miR-21 bidirectional function is relevant in cancer cells as well as in TME and immune cell infiltrate. MiR-21 is the most abundant miRNA expressed by macrophages, endothelial cells and T lymphocytes and its inhibition modulates inflammation depending on TME cell subpopulations. 

A bidirectional activity on HCC and tumor-infiltrating cells was also described for miR-17-92 cluster. MiR-17-92 cluster is a polycistron encoding six oncomiRNAs able to promote proliferation and angiogenesis while inhibiting apoptosis. It takes an active role in the immune system development and differentiation of specific subpopulations, as well as in the establishment and maintenance of B cell lymphoma via a fine regulation of myc oncogene [61]. Indeed, the synergy between c-MYC and miR-17-19b, a truncated version of the miR-17-92 cluster, relevantly contributes to B cell lymphoma initiation as well as to lymphoma cell homeostasis. In line, the contribution of myc-driven signaling is well-established in HCC. Remarkably, the miR-17-92 cluster is highly expressed in a variety of cancers including HCC [62], uncovering a cross-functional role in carcinogenesis. Neoangiogenesis is a key factor enabling HCC development and progression, and it actually represents a pillar diagnostic element and a relevant therapeutic target. In HCC, the miR-17-92 cluster is tightly regulated by VEGF and its upregulation contributes to the angiogenic switch of endothelial cells. By investigating in vitro and in vivo settings, VEGF was shown to trigger miR-17-92 cluster expression in endothelial cells, contributing to endothelial proliferation and angiogenic sprouting, via Erk/Elk1 and PTEN pathways [63]. To further outline the relevance of this miRNA cluster in cancer, it should be recalled that Pten, myc, BIM, E2F1, MMP3, TGFbR2, smad7, and E-cadherin are oncogenic drivers directly targeted by miR-17-92 cluster components, as reviewed by Tan W et al. [64]. These molecules are drivers in HCC, and, at the same time, they are involved in the immune response modulation. Indeed, *miR-17-92* participates in the regulation of the T lymphocytes’ effector response and surveillance against cancer cells. MiR-17-92 ectopic expression in T cells of transgenic mice improves type 1 response, confirming the relevance of this cluster in T cell compartment, and especially in enhancing the Th1-mediated adaptive immunity against cancer [65]. This study also reported a downregulation of the miR-17-92 cluster in tumor-bearing mice and glioblastoma patients and demonstrated that type-2-skewing tumor microenvironment inhibits miR-17-92 expression in T cells and that Th1 helper cells display higher miR-17-92 expression when compared to Th2 cells. In agreement, an increased IFN-gamma production was observed in CD8+ T cells from miR-17-92 transgenic mice [66]. These findings led to the hypothesis that miR-17-92-enforced expression in immune cells might be explored to enhance cancer immunotherapy. MiR-17-92 cluster stimulates T cell proliferation, inhibits T cell death after antigen stimulation, promotes IFN-gamma synthesis, inhibits Tregs’ differentiation and reduces the differentiation of CD8+ memory T cells by targeting BIM, PTEN, TGFβRII and CREB1 [67]. All these relevant immune-modulating functions, especially in the T cell compartment, are intertwined with the above-mentioned oncogenic functions. Of note, miR-17-92 cluster is highly enriched in exosomes secreted by cancer as well as stromal, mesenchymal, and stellate cells [68,69,70] providing a crosstalk function between cancer and TME cells.

To sum up, here we presented examples of crucial oncogenic miRNAs with a pivotal function in the immune response too. The modulatory functions of these miRNAs affect at the same time cancer cells, TME and immune cells (Figure 1). Due to this multifaceted effect of bidirectional miRNAs, we believe that caution should be taken when considering an miRNA-based therapeutic application because contradictory antitumor or pro-tumorigenic effects could be observed. Additional in vivo studies with immunocompetent animal models mirroring the human disease should be carried out for a deeper evaluation of miRNA mimics or antagomiRs as therapeutic agents for an immune-mediated anticancer strategy.

**Figure 1 biomolecules-12-00930-f001:**
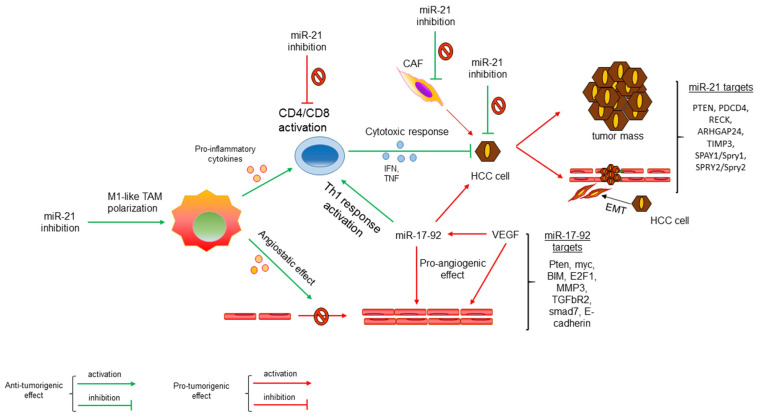
**‘Bi-directional’ effect of deregulated microRNAs in HCC.** MiR-21 and miR-17-92 cluster regulate both HCC phenotype and immune system activation or repression. Their pro- or antitumorigenic effects depend on the regulation of different immune cell subpopulations.

## 5. MicroRNAs Participate in the Modulation of Immune Checkpoint Molecules

The majority of clinical trials in advanced HCC focused on programmed cell death protein 1 (PD-1) and *programmed death-ligand 1* (PD-L1). PD-L1 inhibition associated with bevacizumab has recently received approval for first-line treatment, as a result of the favorable effects of the recent clinical trials. Other immune checkpoints such as CTL4 and LAG3 are expected to gain relevance as well; however, the molecular mechanisms regulating their expression in HCC are still poorly understood. Thus, we will focus primarily on PD-L1 and PD-1 regulation by miRNAs in HCC. Indeed, the immunomodulatory functions of miRNAs in different immune system subpopulations as well as their crosstalk with HCC-specific pathways has been described elsewhere [71], but a systematic analysis of miRNAs directly targeting immune checkpoint molecules was missing.

By binding to PD-1 expressed in immune cells, PD-L1 and PD-L2 inactivate T cells and dampen the immune system reactions against cancer cells. PD-L1 is aberrantly expressed by cancer cells due to multiple genetic and epigenetic events, only partially characterized. Understanding the effective role of mechanisms leading to tumor immune escape in different patient subgroups and characterizing the complex interplay between immunoediting and oncogenic pathways might help to tailor combined treatments to increase ICPIs’ effectiveness. PD-L1 overexpression, revealed by IHC assays, represents an informative biomarker in several cancer types to identify patients likely to receive a benefit from its blockade by monoclonal-antibodies-based therapies. This does not apply to HCC, where no biomarker helps patient selection for ICPI choice. Among the mechanisms sustaining aberrant PD-L1 expression in HCC, direct and indirect roles of miRNAs were reported and some of the most relevant will be detailed below. 

## 6. MicroRNAs Directly Targeting PD-L1

PD-L1, also referred to as CD279 and B7-H1, is a transmembrane glycoprotein mainly expressed by macrophages, activated T and B cells and dendritic cells. Beside inflammatory conditions, PD-L1 expression is mostly elicited in tumor cells as an “adaptive” response to skip the immune surveillance. When expressed by tumor cells, PD-L1 inhibits the anticancer immune response by reducing the proliferation and function of T cells via PD-1 binding that triggers cell death of cytotoxic T cells. 

PD-L1 is a direct target of several miRNAs, most of which were previously shown to modulate oncogenes or tumor suppressor genes. Among them, **miR-378a-3p**, downregulated in two HCC datasets, directly targets PD-L1 and induces apoptosis by repressing STAT3 signaling in HCC cells [72]. MiR-378a-3p showed a non-redundant function through acting on both PD-L1 and STAT3. This last event was responsible for reduced proliferation, migration and invasion suggesting miR-378a-3p restoration as a possible strategy to hint cancer hallmarks and favor anticancer immune response by inhibiting immune escape via PD-L1 downregulation and Treg inhibition. Interestingly, the enforced expression of miR-378a-3p in HCC cell lines caused PD-L1 silencing and consequently reduced Tregs’ induction in co-culture assays. 

PD-L1 upregulation in HCC cells can be driven also by the EGFR-P38 MAPK axis, via **miR-675-5p** [73]. Interestingly, phospho-EGFR directly correlated with PD-L1 expression and inversely correlated with HLA-ABC in HCC tissues and cell lines. The molecular mechanisms underlying these alterations started from an activation of P38-MAPK responsible for both miR-675-5p downregulation and hexokinase2 (HK2) upregulation. In turn, miR-675-5p downregulation led to an accumulation of PD-L1 via mRNA stabilization, while HK2 upregulation increased aerobic glycolysis and reduced HLA-ABC expression. Increased PD-L1-mediated immune suppression and reduced HLA-ABC expression impaired T-cell-mediated lysis of cancer cells as well as antigen-presenting ability, and cooperated to promote immune evasion, and more aggressive clinicopathological features [74,75]. This immune-regulatory and metabolic pathway intricacy confirms the role of miRNAs as modulators of multiple events contributing to the development of the cancer phenotype and tumor progression. Consequently, in a therapeutic perspective, miRNA modulation might represent an appealing multi-target approach in combination with immunotherapy. 

Experimental evidence on immune checkpoint targeting by miRNAs is limited when liver cancer is considered; however, many miRNAs deregulated in HCC have been shown to target these molecules in other cancer types. These findings cannot be translated to HCC, as we know that miRNA functions are strongly dependent on the molecular context. Notwithstanding, we will briefly report data on miRNAs deregulated in HCC which were demonstrated to target PD-L1 or PD-1, either in HCC or in other cancer types, since it is conceivable that they might contribute to the deregulated expression of immune checkpoints in HCC as well.

The **miR-142** stem loop structure encodes for miR142–5p and miR142–3p. MiR-142 is highly expressed in hematopoietic cells, participating in lineage differentiation and modulation of the immune response. Indeed, **MiR-142** target genes are enriched in pathways involved in the regulation of the immune response as well as in several immune-related diseases [76]. In pancreatic cancer cells, miR-142-5p inhibits PD-L1 expression and suppressed in vivo tumor growth, suggesting its reinforced expression to potentiate ICPIs treatment [77]. Interestingly, miR-142-5p influences the tumor microenvironment, PD-L1 expression and HCC prognosis by targeting m^6^A RNA methylation regulators [78], warranting its validation in advanced HCC undergoing immunotherapy. Beside immunoediting functions, miR-142 also plays a relevant oncogenic role. MiR-142-3p is a TS miRNA, the downregulation of which in tumor tissue has been advocated as responsible for a multifaceted oncogenic action in HCC. It facilitates the invasion and migration of HCC cells by targeting high-mobility group box protein 1 [79]. Similarly, miR-142-5p-enforced expression reduces cell viability and promotes apoptosis in HCC cell lines via FOXO (Forkhead box, class 0, 1 and 3), Bim (a Bcl-2-interacting mediator of cell death), procaspase 3, and activated caspase 3 targeting, thus acting on both cell viability and apoptosis [80]. MiR-142-5p downregulation is associated with a worse prognosis in HCC patients and its enforced expression inhibits HCC cell migration [81]. Furthermore, miR-142-3p-reduced expression in HCC cells gives them a metabolic advantage by directly inhibiting LDHA (lactate dehydrogenase A), favoring aerobic glycolysis and proliferation of cells that have undergone metabolic reprogramming [82]. To sum up, miR-142 mature isoforms contribute to immunoediting, metabolic and oncogenic functions, representing a further example of the multi-directional functions of miRNAs as drivers of cancer development and progression. Interestingly, miR-142-3p downregulation results from the deranged expression of a lncRNA, namely MALAT1 [83]. MALAT1 is upregulated in HCC, and in vitro findings suggest that by sponging miR-142-3p, it enhances tumor growth, cell proliferation, migration and invasion. Similarly, TUG1 (taurine-upregulated gene 1) lncRNA, upregulated in HCC, reduces miR-142-3p levels acting as a competitor endogenous RNA [84]. In turn, miR-142-3p downregulation is responsible for ZEB1 overexpression, demonstrating the TUG1/miR-142-3p/ZEB1 axis as a driver event in HCC cells.

MiR-570 is the most deregulated miRNA in the progression of alcoholic liver disease to HCC. Twenty-six genes are deregulated because of miR-570 downregulation and contribute to cell proliferation and the invasion capability of HCC cells [85]. Among miR-570 direct targets, PD-L1 was reported to play a crucial role not only in immunoediting, but also in the determination of cancer hallmarks such as proliferation and invasion [86]. Indeed, rescue experiments in miR-570 overexpressing cells confirmed that PD-L1 restoration reversed the inhibition of proliferation and invasion induced by miR-570-enforced expression, outlying the dual oncogenic and immunoediting role of this molecule. These in vitro findings were also confirmed in vivo, showing miR-570-driven suppression of tumorigenicity and metastasis. In a xenograft model of HCC-derived SMMC7721 cells, its restored expression was able to trigger apoptosis and to inhibit angiogenesis, as confirmed by CD31 and VEGF inhibition. Moreover, miR-570 overexpression promoted T cell activation and proliferation, as shown by a lower ratio of CD3+ CD4+ T cells and higher ratio of CD8+ IFN-γ+ T cells in peripheral blood and tumor tissues of miR-570 mimics-treated mice [87]. Thus, miR-570 displays a dual-tumor-suppressive and immune-eliciting role and its downregulation in HCC contributes to immune escape, tumor growth and reduced apoptosis. 

To conclude the series of representative miRNAs playing dual roles in tumorigenesis and immune response modulation, we need to mention **miR-122**. It is the most abundant miRNA expressed by hepatocytes, accounting for 70% of miRNA liver population. Its downregulation in HCC cells was linked to tumor development via targeting of crucial oncogenes such as cyclin G1 [45], ADAM10, IGF-1 receptor, ADAM17, CUTL1, Pkm2, Wnt1, pituitary-tumor-transforming gene 1 binding factor, Cut-like homeobox 1, c-myc, bcl-w, wnt-1, and others, as reviewed by Nakao et al. [88]. Besides the oncogenic contribution of deregulated miR-122 to HCC aggressiveness, recent investigations have revealed its role in the modulation of the immune response. In the setting of innate immunity, miR-122 directly targets TLR4 [89] which, in turn, triggers proliferation [90] and drug resistance of HCC cells through the COX-2/PGE2/STAT3 positive feedback loop [91]. Remarkably, TLR4 is expressed in HCC cells, Kupffer cells, activated stellate cells, and actively participates in multiple pathways involved in the innate immune response as described by Shi L. et al. [89], who also showed that miR-122 mimics strongly reduce TLR4 downstream cytokines, IL-6 and TNF-α. These findings outline the dual function of miR-122 in HCC, acting as both an oncogenic factor and modulator of the innate immunity (Figure 2). 

**Figure 2 biomolecules-12-00930-f002:**
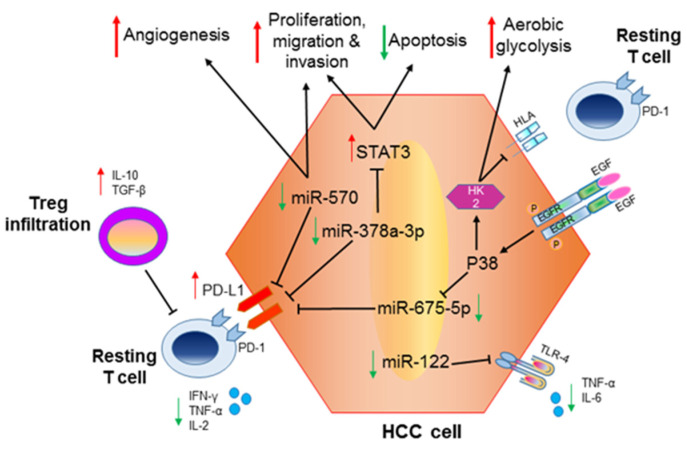
**HCC-deregulated microRNAs modulate both PD-L1 expression and tumorigenic pathways.** Contemporaneous regulation of oncogenic genes and PD-L1 expression promotes HCC aggressiveness and immune escape of tumor cells. Green arrows: downregulated miRNAs/genes; red arrows upregulated miRNAs/genes.

## 7. MicroRNAs Directly Targeting PD-1

PD-1 regulates T cell function and inhibits adaptive and innate immune response, while reversing immune tolerance when silenced. PD-1-induced suppression of T cells is mainly mediated by the activation of apoptotic cell death. Several lineages of immune cells express PD-1, such as activated T cells and in particular tumor-specific T cells, natural killer, B lymphocytes, macrophages, dendritic cells (DCs) and monocytes. In HCC, infiltrating T cells are functionally inhibited by multiple immune suppression mechanisms, such as the PD-1 pathway that diminishes antitumor immunity, impairing T cell proliferation and cytotoxicity, and IFN-γ production and release in the TME. Clinical trials have shown that antibody-mediated blockade of PD-1 may induce the regression of several tumor types, including a subgroup of HCCs [92]. Some downregulated miRNAs are responsible for PD-1 overexpression, contributing to immunotolerance in HCC (Table 2).

**MiR-15a-5p** is downregulated in HCC tissue and cell lines, and its low expression is associated with poor prognosis. MiR-15a-5p is packaged into exosomes by cancer cells and is delivered to CD8+ T cells where it inhibits PD-1 by direct targeting [93].

**MiR-138** is downregulated in HCC tissue and in particular in HBV-related cases [94]. In agreement, it inversely correlated with HBV viral load in patients affected by liver diseases, where it directly targets PD-1 in primary CD3+ T cells [95]. These authors suggested that miR-138 promotes T cell response by inducing PD-1 blockade in HBV-infected patients, outlying its restoration as a possible therapeutic approach. Indeed, miR-138-enforced expression in primary T cells inhibited PD-1 and triggered cytokine secretion. MiR-138 activity on immune system players was investigated in several cancer types. In colorectal cancer, it directly inhibits PD-L1 [96], in NSCLC it directly targets PD-L1 and PD-1 [97], and in glioblastoma it downregulates both PD-1 and CTLA4 expression [98], whose relevance in HCC is well-known. In addition to PD-1 inhibition, miR-138 modulates the expression of other oncogenic drivers in HCC, such as cyclin D3 [99], SOX9 [100], and SIRT1 [101], confirming its dual function in oncogenic and immunoediting processes. Although not directly obtained in HCC, we can suppose that the frequent miR-138 downregulation might result in CTLA4, PD-1 and PD-L1 overexpression, all converging on immuno-tolerance of liver cancer. 

**MiR-374b** was shown to target PD-1 in HCC and its enforced expression improved the antitumor potential of cytokine-induced killer cells [102].

To add a further layer of complexity, it was also shown that allele polymorphisms might strongly affect the effectiveness of miRNA-induced gene expression modulation. As an example, **miR-4717** inhibited PD-1 expression in lymphocytes from chronic HBV patients bearing the rs10204525 polymorphic site of allele GG. Conversely, miR-4717 did not display any effect on the rs10204525 genotype AA, demonstrating that this miRNA regulates PD-1 expression in an allele-specific way, thus differentially influencing the immune response and HBV disease course according to the presence of specific SNPs [103]. Thus, miR-4717 allele specificity towards the PD-1 gene may explain, at least in part, the variability in HBV infection, the different patterns of cytokines, and PD-1-inhibitory effect on T cell activation of HBV-infected patients [104,105], ultimately leading to the heterogeneous evolution of HBV-related chronic liver disease.

Here, we reported some miRNAs consistently deregulated in HCC, which were demonstrated to directly target immune checkpoints. In a few cases, the direct targeting was proven in other cancers; however, strong evidence supports their relevant function in HCC too. Even though an extrapolation among different cancer types cannot be sustained in the absence of experimental investigations, we tried to give a more general vision of the outstanding role that miRNAs play in the immunoediting process. We missed reporting the indirect mechanisms leading to immune checkpoints’ deregulation because their complexity needs to be verified in each tumor type. The same applies to other immune molecules, such as CTLA4 and LAG3, whose expression is also relevant in HCC. However, miRNA-dependent mechanisms driving aberrant protein levels in this cancer type are still poorly understood. 

CTLA4 is expressed by T-regulatory cells and inhibits cytotoxic response against malignancies. Fayyad-Kazan et al. studied the miRNA signature of circulating CD4 Treg lymphocytes in healthy subjects, and found a group of miRNAs with direct and indirect regulatory functions on FOXP3 and CTLA4 that are critical for Treg activities. More in detail, this signature is characterized by miR-95 and miR-509 overexpression and by miR-9, -18a, -24, -27b, -126, -133a, -134, -145, -181b, -181d, -210, -224, and -335 downregulation and differentiates CD4+ CD25+ CD127low Treg from CD4+ CD25– peripheral T cells. Nevertheless, its identification in healthy volunteers might not reflect the immune system in the presence of malignant diseases [106]. Remarkably, among these miRNAs, miR-24 and miR-210 inhibited Foxp3 expression, while miR-145 inhibited CTLA4 by binding to its 3-UTR, paving the way to possible combined therapeutic approaches by restoring miR-145 levels to enhance the cytotoxic response.

**Table 2 biomolecules-12-00930-t002:** MicroRNAs targeting PD-1 in liver diseases.

miRNA	miRNA Expression	Liver Disease	Target Cells	Preclinical Models	Reference
miR-15a-5p	downregulated	HCC	CD8+ T cells	HepG2 cells	[93]
**miR-138**	downregulated	HCC, CHB ^1^, LC ^2^	CD3+ T cells	Primary lymphocytes	[95]
**miR-374b**	downregulated	HCC	Cytokine-induced killer cells	HepG2 cells, xenograft mouse	[102]
**miR-4717**	downregulated	CHB ^1^	Peripheral lymphocytes	HepG2 cells	[103]

^1^ CHB: chronic hepatitis B; ^2^ LC: liver cirrhosis.

## 8. Conclusions and Clinical Significance

In this review, we introduced the role of some selected miRNAs in the immune system development as well as in B and T cell program activation. Then, we focused on miRNAs’ modulation of immune checkpoints, PD-L1 and PD-1, highlighting their role in immune tolerance in HCC. We reported the dual role of tumor-specific miRNAs as regulators of PD-1/PD-L1 as well as oncogenic drivers, emphasizing how these miRNAs could serve as both biomarkers and putative therapeutic targets. The bidirectional functions of such miRNAs represent a still poorly investigated field with very few validated results in preclinical studies. Nevertheless, if confirmed in preclinical models of HCC (animal models and co-culture assays), these findings might represent preliminary evidence to be further investigated in the setting of ICPI treatments, to identify possible predictive biomarkers helping patient allocation to personalized treatments. Indeed, the allocation of patients to ICPI-based treatments, the choice of specific ICPIs’ associations or the choice of ICPI plus anti-VEGF treatment is not guided by any biomarker. The same applies to biomarkers informative for early and late resistance to such treatments, which are still lacking. Interestingly, if these microRNAs should be recognized as informative biomarkers, they might also represent hypothetical therapeutic targets to be tested for the development of combined immunotherapy strategies in HCC. 

HCC is an inflammation-linked tumor which develops in a highly immunotolerant organ. This premise supports the likely effectiveness of immunotherapy in the treatment of this disease. The marked molecular heterogeneity of HCC may account for the difficulty in the identification of biomarkers helping patient allocation to the best therapeutic option. Knowing the molecular events driving HCC development, immune escape and immunoediting represents the mandatory step to tailor the most effective approach. 

MiRNAs take part in immune system development and in the activation of specific programs leading to cancer development, cancer-induced immunoediting, and the modulation of immune response against tumors. Indeed, miRNAs participate in B-cell differentiation and function and, even more interestingly, when dealing with cancer-directed immune response, their impaired expression determines a reduction in thymic and peripheral T cells. In addition, miRNAs’ inhibition resulting from DICER inactivation determines a Th1 and Treg induction and impairs Th17 levels. Thymic-derived and peripheral Foxp3-dependent Tregs strongly depend on miRNAs for their number and functional program, as revealed by Dicer or Drosha ablation. Tregs are enriched in the infiltrate of many human cancers thus favoring the immune escape of tumor cells via the inhibition of effector T cells [107]. In addition, PD-L1 upregulation correlates with Tregs in the tumor microenvironment. Moreover, many miRNAs have been recognized as modulators of immune checkpoints such as PD-L1, PD-1 and CTLA4. It is striking that many miRNAs acting on the immune system and immune response regulation also target oncogenic drivers involved in redundant and parallel pathways. Even though specific targets seem to drive miRNA effects in each context, it might be speculated that miRNA activity on target mRNAs is likely to be cumulative; thus, in “real life”, the role of miRNAs is expected to be more relevant and complex than single events explored in laboratory settings. These data let us suppose that in the case of cancer driver and immune-modulating ‘dual’ miRNAs, a therapeutic exploitation might be envisaged. Unravelling miRNAs’ role and dissecting their specific functions to use them as biomarkers or therapeutic tools remains difficult due to the intricacy of miRNA networks, regulatory loops in which they are embedded, and conflicting effects depending on molecular and genetic contexts. This complexity has necessarily slowed down the dissection of their physiopathologic functions and the identification of their possible therapeutic potential. In this scenario, the study of bidirectional miRNAs adds a further layer of complexity to preclinical investigations due to the multicellular effects of a single miRNA on both tumor cells and TME subpopulations, requiring the employment of immunocompetent animal models and 2D/3D co-culture assays to fully validate these promising results. 

The ultimate reason for clarifying molecular mechanisms regulating immune checkpoints’ aberrant expression is to identify possible biomarkers to select the patients who will benefit from ICPI administration and to identify possible combined treatments to potentiate their antitumor effects. Remarkably, cancer immunoediting is a continuously evolving process in the course of malignant diseases, which is also amplified by immune-related treatments and targeted therapies. Immunoediting modulates cancer cell immunogenicity to skip the immune response in favor of immune-driven tumor clone selection, which is at the basis of immune escape, disease progression and resistance to treatments. Whether miRNAs might be combined with immunomodulating treatments aiming to inhibit immunoediting and oncogenic drivers at once still needs to be carefully verified. However, the preclinical findings collected so far seem to argue in favor of this hypothesis due to the effective modulation of molecular pathways driving oncogenic and immune-skipping phenotypes associated with cancer aggressiveness. Although promising, these data need to be further investigated in specific HCC subtypes and representative animal models before being accepted in order to avoid unattended opposing results depending on the target cell population. Circulating miRNAs have not been investigated in this specific setting. Conversely, circulating miRNA panels were shown to have an accurate diagnostic performance for HCC [108] and they were suggested to have a prognostic and predictive role too [109]. 

Unfortunately, circulating miRNAs associated with “hot” and “cold” HCCs, based on the activation or inhibition of the immune response, have not been reported so far. This represents a very interesting field of investigation that might provide not only a source of biomarkers to be tested as putative predictive tools in the setting of immunotherapy, but also a dynamic monitoring of changes in the tumor–immune system interplay deriving from disease progression or resulting from treatments.

To sum up, the multifaceted spectrum of targets allows miRNAs to modulate at the same time immune-associated factors and oncogenic or TS drivers. Since miRNAs are pivotal in maintaining homeostasis, it is intuitive that their derangement might contribute to disease outbreak by acting at multiple levels. Their manipulation may not be strong enough to be proposed as a monotherapy, yet they might be investigated in combined approaches to dissect their effectiveness in strengthening the response to treatments and prevent escape mechanisms, due to their impact on multiple redundant and parallel-signaling cascades.

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
