# Peer review of "MicroRNAs at the Crossroad between Immunoediting and Oncogenic Drivers in Hepatocellular Carcinoma"

_biomolecules, 2022, doi:10.3390/biom12070930_

Round 1

Reviewer 1 Report

In this review, Gramantieri et al tried to summarize the current understanding about the role of miRNA in hepatocelluar cancer (HCC). But this review is not well organized, it's just a plain listing of current publications, there is not much writer's thought in this review. Also, there is too much descriptions about the basic tumour immunology information. 

I believe this review needs some improvement. The authors have to make the main message clearer and providing some new thoughts from the current publication.

Author Response

In this review, Gramantieri et al tried to summarize the current understanding about the role of miRNA in hepatocelluar cancer (HCC).

- Answer: this not the exact aim of our review. We apologize for the lack of clearness of the real aim of the review. In the amended version of the review we have defined more clearly our aim to focus specifically on bidirectional miRNAs, influencing both oncogenic and immune-related pathways at the same time. Additional sentences have been added at the end of the summary, as well as at the end of the introduction.

But this review is not well organized, it's just a plain listing of current publications, there is not much writer's thought in this review. Also, there is too much descriptions about the basic tumour immunology information. I believe this review needs some improvement. The authors have to make the main message clearer and providing some new thoughts from the current publication.

- Answer: The subject of this review (the bidirectional microRNAs acting on both oncogenic and immune-modulating pathways) is still poorly explored so far in HCC. In our opinion, this represents the originality of this review. Many other reviews deal with the role of microRNAs in HCC and we didn’t mean to duplicate these. We think that the role played by miRNAs in the interplay between oncogenic and immune mediated pathways is an interesting subject and we wanted to focus on it, also to trigger the interest towards this still poorly explored subject.

We have added some more personal insights into the reviewed data (see tracked version). We have however outlined how this field is still poorly investigated. Indeed, validated findings are really few, thus we still need to be cautious before drawing personal conclusions. Two more sentences have been added in the conclusions, dealing with the lack of validated investigations on this subject and the lack of studies on circulating miRNAs modulating the immune system.

We have reduced the description of basic tumor immunology as requested by the Reviewer.

Reviewer 2 Report

The manuscript is well written, the authors flow from one idea to another and the figures presented are well developed. However, I have some concerns:

1- The authors should highlight the novel findings or aspects in their manuscript and show the differences between their manuscript and recently published as PMID: 34049801  DOI: 10.1016/j.semcdb.2021.05.025

2- The author should also add a table to summarize the selected miRNAs showing upregulation/downregulation, model, target genes and relation to immunomodulation.

3- I think the authors missed to cite important work about the expression profile of  miR-122, members of miR-17-92 cluster e.g., miR-19a, and other  miRNAs related to HCC and inflammatory processes in liver fibrosis such as  PMID: 26352740PMID: 28211229

4- I think adding a title that highlights the role of selected miRNAs as circulating biomarkers in HCC and another part that emphasize the clinical relevance of selected miRNAs in personalized medicine and as therapeutic targets will strengthen this review.

5-The font size should be adjusted to be the same throughout the manuscript.

  •  
  •  
  •  

Author Response

The manuscript is well written, the authors flow from one idea to another and the figures presented are well developed. However, I have some concerns:

1- The authors should highlight the novel findings or aspects in their manuscript and show the differences between their manuscript and recently published as PMID: 34049801  DOI: 10.1016/j.semcdb.2021.05.025

Answer:

At the end of the summary, as well as at the end of the introduction we have defined more clearly our aim to focus specifically on bidirectional miRNAs, influencing both oncogenic and immune-related pathways at the same time. This represents the most relevant difference with respect to the major part of previously published reviews.

In the chapter entitled ‘MicroRNAs participate to the modulation of immune checkpoint molecules’ we have added a sentence detailing the differences between our manuscript and the recently published review PMID: 34049801 (Ref. 71). In that review, the Authors consider and describe the immunomodulatory functions of miRNAs in different immune system subpopulations as well as their crosstalk with HCC-specific pathways but do not perform a systematic analysis of miRNAs directly targeting immune checkpoint molecules (PD-1 and PD-L1). In addition, the Authors consider other examples of bifunctional miRNAs (e.g., miR-615-5p) that are different from those detailed in our manuscript (e.g., miR-17-92 and miR-21) and they do not emphasize this point which we consider a crucial aspect for further investigations for combined immunotherapy and miRNA-based strategies in HCC.

2- The author should also add a table to summarize the selected miRNAs showing upregulation/downregulation, model, target genes and relation to immunomodulation.

Answer: As requested we have added two new tables. Table 1 summarizes the effects of miRNAs on B and T cells maturation and activation processes, describing target genes, immunologic effect, and miRNA levels. Table 2 lists miRNAs involved in PD-1 regulation in HCC describing the kind of liver disease, the target cells and the preclinical models. We did not add other tables on bidirectional miRNAs or on miRNAs participating to the modulation of immune checkpoints because the two Figures already summarize these aspects.

3- I think the authors missed to cite important work about the expression profile of  miR-122, members of miR-17-92 cluster e.g., miR-19a, and other  miRNAs related to HCC and inflammatory processes in liver fibrosis such as  PMID: 26352740 , PMID: 28211229

Answer: Regarding the suggested paper PMID: 26352740 (Ref. N. 49), we have added a sentence commenting the role of immune modulatory miRNAs (e.g., miR-19a) as possible diagnostic biomarkers of early HCC (page 14). Moreover, we added two further studies (Ref. N. 50, 51) regarding the immune regulatory role of exosome-associated miRNAs. We therefore changed the chapter title ‘MicroRNAs in the immune response against HCC’ as follow: ‘MicroRNAs as immune modulatory molecules mediating cell-cell crosstalk in HCC’.

On the contrary, we decided not to mention the other suggested paper (PMID: 28211229) because our review is focused on HCC and not on fibrosis and we would like to avoid to go off topic.

4- I think adding a title that highlights the role of selected miRNAs as circulating biomarkers in HCC and another part that emphasize the clinical relevance of selected miRNAs in personalized medicine and as therapeutic targets will strengthen this review.

Answer: Unfortunately, the field of bi-directional microRNAs playing a role in both oncogenic and immune-mediated pathways is a still poorly explored field with really few validated data. In addition, findings on this subject come almost exclusively from tissue studies. Conversely, circulating miRNAs and circulating miRNA panels were investigated in different settings, with the aim to identify diagnostic, prognostic or predictive biomarkers. Therefore, introducing these data, which are outside the subject of this review, appears to be misleading. As suggested, we modified the title of the last chapter to ‘Conclusion and clinical relevance’ and added few sentences dealing with the clinical potential of the findings presented in this review.  We also highlighted the lack of validated investigations on bi-directional miRNAs and the lack of studies on circulating miRNAs modulating the immune system.

5-The font size should be adjusted to be the same throughout the manuscript.

Answer: The font size has been adjusted throughout the manuscript.

Reviewer 3 Report

The present manuscript reviewed the biological roles of miRNAs in hepatocellular carcinoma (HCC) especially on the mechanism of various immunological phenomena. The reviewer considers that the present manuscript has a sufficient number of contents. The reviewer would request some queries as described below.

1.    The reviewer considers that several references need to be added in the sentences of line 351-360 and line 471-473.

2.    There are many typographical errors in the main text. The authors should proofread the main text carefully.

 In addition, the reviewer considers that the conclusion of the present manuscript is too long.

Author Response

The present manuscript reviewed the biological roles of miRNAs in hepatocellular carcinoma (HCC) especially on the mechanism of various immunological phenomena. The reviewer considers that the present manuscript has a sufficient number of contents. The reviewer would request some queries as described below.

  1. The reviewer considers that several references need to be added in the sentences of line 351-360 and line 471-473.

Answer: We agree with the Reviewer that few references were missed in the indicated paragraphs. The following References (Ref. N. 65-67) were added in the revised version of our manuscript (old paragraph 351-360). Regarding the sentence at lines 471-473, we moved above the reference N. 86 (old Ref. N. 79) which makes the reference association clearer. We thank the Reviewer for this specific comment.

  1. There are many typographical errors in the main text. The authors should proofread the main text carefully.

Answer: Typographical errors have been corrected by a careful proofreading of the authors. We thank the Reviewer for this elucidation.

Round 2

Reviewer 1 Report

I think this manucript is ready for publication.

Reviewer 2 Report

The authors have changed as required.